# Supernumerary Extraocular Muscle: A Rare Cause of Atypical Restrictive Strabismus

**DOI:** 10.3390/medicina58111691

**Published:** 2022-11-21

**Authors:** Xiangjun Wang, Tao Shen, Mengya Han, Jianhua Yan

**Affiliations:** State Key Laboratory of Ophthalmology, Zhongshan Ophthalmic Center, Sun Yat-sen University, Guangdong Provincial Key Laboratory of Ophthalmology and Visual Science, Guangdong Provincial Clinical Research Center for Ocular Diseases, Guangzhou 510060, China

**Keywords:** supernumerary extraocular muscle, restrictive strabismus, strabismus surgery

## Abstract

*Background and objectives*: Supernumerary extraocular muscle (SEOM) is extremely rare. The purpose of this paper was to review the clinical characteristics and surgical outcomes of SEOM patients with atypical restrictive strabismus. *Materials and Methods*: A retrospective review was conducted on the data from 12 SEOM cases. Pre- and post-operative measurements consisted of visual acuity, cycloplegic refraction, ocular alignment, ocular motility, binocular vision, and imaging. Management strategies included either conservative or surgical treatments. *Results*: Of the 12 cases reviewed (seven females, five males), the mean ± SD age was 14.3 ± 10.6 years (range: 4–38 years). The right eye was affected in six cases, the left in five, and both eyes in one case. The major clinical manifestations included restrictive ocular motility (12 cases), with seven cases in no less than three directions; varying degrees of horizontal or vertical strabismus; ipsilateral amblyopia (10 cases); and unequal palpebral aperture (10 cases). Imaging results revealed muscular bands originating from the annulus of Zinn and insertion into the globe or other recti, as well as anomalous muscular bands connecting two or more recti, sometimes with optic nerve involvement. Three patients received conservative treatment, while rectus recession with or without resection (seven patients) or rectus disinsertion plus globe fixation (two patients) were performed in those receiving surgical treatments. A surgical success was achieved in four cases. *Conclusions*: For restrictive strabismus, imaging plays an important role in the diagnosis of SEOM. When the SEOM is difficult to resect, a personalized surgical strategy may be required to achieve a good ocular alignment.

## 1. Introduction

Supernumerary extraocular muscle (SEOM) is an extremely uncommon condition in which more than six extraocular muscles (EOMs) are present, and it can result in restrictive motility and strabismus. Prior to precise imaging diagnostic techniques, SEOM was often only diagnosed at autopsy or during strabismus surgery. The first case of SEOM was reported by Nussbaumin in 1893 and involved an abnormal portion of tissue projecting from the lateral rectus to other recti [1,2,3]. Subsequently, other cases observed in outpatient settings have been reported [4,5,6]. For example, Valmaggia et al. [5] described a case with a fusiform accessory intraconal muscle coursing from the annulus of Zinn and inserting into the posterior sclera of the globe. In spite of an elevation deficiency of the left eye, this patient showed orthophoria in the primary position and had good binocular vision. Lueder [7] classified these anomalous anatomic variations into three types: (1) the abnormal insertion of the accessory muscle originating from a normal EOM, (2) fibrous bands under the rectus muscle, and (3) a supernumerary muscle coursing from the posterior orbit, with some of these cases confirmed to be SEOM based on pathological examinations.

The embryology and etiology of SEOM remain unclear; however, results from previous studies suggest that these anomalies could be an atavistic situation. Support for this hypothesis follows from the presence of an oculomotor innervated retractor bulbi muscle located along the rectus muscles in many vertebrates, which originates from the annulus of Zinn to traverse anteriorly and adhere to the posterior part of the globe. An alternative hypothesis is that SEOM arises due to a disorder during early developmental stages rather than a retractor muscle [2,8].

Most of these patients were initially misdiagnosed as having congenital fibrosis of the extraocular muscle (CFEOM) or Duane syndrome when first seen at various outpatient centers. However, the imaging findings revealed that these patients showed a SEOM within the involved orbit. With the development of high-resolution magnetic resonance imaging (MRI), the recognition of such abnormalities and a more accurate characterization regarding the location and shape of EOMs has become possible [9]. Khitri and Demer [10] first analyzed 453 strabismus subjects and reported that the probability of SEOM in cases of strabismus was 2.4%, which is not as rare as once thought. In the same study, the authors observed that most supernumerary bands were smaller than normal EOMs and were either located between two muscles or inserted into the globe.

Overall, our knowledge of SEOM remains quite limited due to its low incidence and sparse reports in the literature. As a result, patients with SEOM may tend to be overlooked by ophthalmologists and then undergo inappropriate treatments or even multiple surgeries without a satisfactory outcome. Here, we present a case series of 12 SEOM patients with atypical restrictive strabismus and review the clinical characteristics, imaging findings, and final surgical outcomes.

## 2. Patients and Methods

This retrospective study was conducted at the Zhongshan Ophthalmic Center of Sun Yat-sen University, China. A detailed review of the records of 12 patients presenting with restrictive strabismus and SEOM, based on imaging, was conducted over the period from July 2011 to November 2020. The inclusion criteria for these cases consisted of the following clinical manifestations: (1) limited ocular motility, (2) horizontal or vertical strabismus, and (3) SEOM in the orbit or between recti according to results of preoperative imaging [7,10]. Informed consent was obtained from all the patients and their parents. Ethical permission for the present investigation was obtained through the Research Ethics Board of the Zhongshan Ophthalmic Center, Sun Yat-sen University, China.

The medical history included age, gender, subjective symptoms, previous treatment, and other ocular or systemic diseases. Pre- and postoperative examinations consisted of visual acuity, cycloplegic refraction, ocular alignment, ocular motility, palpebral aperture size, exophthalmos, and anterior segment and fundus imaging. The ocular alignment in the primary and diagnostic positions was measured using an alternate prism cover test in the 33 cm and 6 m fixation positions. In patients with a very large angle of strabismus who were incapable of maintaining their primary gaze, the perimeter was applied to measure the deviation. Ocular motility was recorded using a diagrammatic representation with a −4 to +4 rating system [11]. Binocular vision at near was evaluated using the Titmus stereoacuity test and at far by random-dot stereograms. Orbital MRI or computerized tomography (CT) scans were performed in each patient. Under topical anesthesia, forced duction and active force generation tests were routinely conducted to determine the restrictive force and muscle contractility.

Conservative treatment (observation) was performed in the outpatient department if the degree of strabismus was ≤10 prism diopters (PD) in primary gaze and there was no abnormal head posture [12], while patients presenting with deviations of >15 PD in the primary gaze received strabismus surgery. Surgical options for the cases in this study depended on the angle of deviation, imaging results, intraoperative assessment, and the possibility of a suspicious anomalous structure excision. Based on the CT or MRI results, a complete resection of the supernumerary EOM bands was the preferred procedure to reduce restrictive elements. In patients with deviations of ≤25 PD, a rectus recession was implemented, while for those with deviations of >25 PD, a rectus recession and ipsilateral antagonist resection were employed. A rectus disinsertion and globe fixation, by suturing silk from muscle insertion into the orbital wall to achieve mild overcorrection immediately after surgery, were used in patients with excessive strabismus where muscle exposure was difficult and it was impossible to perform sutures [13,14].

The principal goal of the surgeries was to achieve orthophoria in the primary gaze. A postoperative ocular deviation in the primary gaze of ≤10 PD in the horizontal and ≤5 PD in the vertical direction was considered as a surgical success. The postoperative follow-ups were conducted at 1 week, 2 months, 6 months, and every year thereafter.

Statistical analysis was performed using SPSS version 26 (SPSS Inc., Chicago, IL, USA). Data from pre- versus postoperative parameters were compared using the Wilcoxon signed-rank test. The means, standard deviations (SDs), and interquartile ranges of the outcomes are presented. This study was determined to be exempt from the Research Ethics Board of the Zhongshan Ophthalmic Center, Sun Yat-sen University, China.

## 3. Results

Among the 12 cases reviewed, there were seven females and five males with a mean ± SD age of 14.3 ± 10.6 years (range: 4–38 years). The right eye was affected in six cases, the left in five cases, and both eyes in one case. Ocular abnormalities were detected during infancy in eight patients and in childhood for the remaining four patients. Most patients experienced limited ocular motility and others strabismus or an unequal palpebral aperture. Case 3 had received a previous surgery with a lateral rectus recession of 9 mm and a medial rectus resection of 14 mm within the right eye (Table 1).

Clinical manifestations varied among the 12 cases; a detailed summary of these manifestations is contained within Table 1. Briefly, some of the clinical characteristics of these 12 cases included: (1) all cases had restrictive ocular motility, with seven cases showing restrictions in multiple (≥3) directions; (2) eight cases presented with horizontal strabismus (>10 PD), nine with vertical deviation (≥5 PD), and five with a very large angle of deviation (>60 PD); (3) the best corrected visual acuity (BCVA) of the affected eye was at least two lines worse than the contralateral eye in 10 cases, with five of these being no more than 2/20; and (4) accompanying eyelid anomalies were present in 11 cases, with the main anomaly being an unequal palpebral aperture in 10 cases (Table 2). In addition, the ocular motility of the contralateral side changed in some cases. The limitation of the affected eye caused slight secondary contralateral yoke muscle overaction in several cases, including the inferior oblique (Cases 8 and 9), superior rectus (Case 8), or superior oblique muscle (Cases 7 and 12). No pre- or postoperative stereopsis was observed in any of the patients. Of the five cases with compensatory head postures, Cases 5, 8, and 12 turned to the left, and an obvious head turn was present in Cases 4 and 9.

The imaging results indicated that the SEOM cases could be classified into four types (Figure 1, Figure 2, Figure 3, Figure 4 and Figure 5): (1) bilateral supernumerary EOM coursing from the anulus of Zinn to the posterior sclera of the globe—Case 1 (Figure 1); (2) connections coursing from the anulus of Zinn and inserting into other extraocular rectus muscles (ERM)—Cases 2, 3, and 4 (Figure 2); (3) connections among ERMs with or without optic nerve (ON) involvement—Cases 5, 6, 7, 8, 9, and 10 (Figure 3); and (4) connections only between rectus and the ON—Cases 11 and 12 (Figure 4).

Of the 12 patients, three were not subjected to any corrective surgery for their mild strabismus in the primary gaze or refused surgery, while the others underwent surgery to alleviate the restriction and correct the strabismus (Table 1). Rectus recessions with or without antagonist resection were performed in seven cases, and for two cases with a very large angle of strabismus (>60 PD), muscle disinsertion and globe fixation were implemented. Among the nine patients receiving surgery, a surgical success was achieved in four cases, as observed at an average follow-up of 28 months. Notably, of the four cases failing to reach orthophoria, Case 6 and Case 10 showed a distinct decrease in horizontal and vertical deviation postoperatively. Ocular alignments as assessed via long-term observations (≥6 months) were consistent with those of short-term outcomes (≤2 months) in six cases, with the exceptions of Cases 1, 7, and 12, who showed deviation angle increases of 20, 25, and 10 PD, respectively. Following surgery, there was a slight improvement in six cases, but no statistically significant changes were obtained in three of these cases (Table 3).

## 4. Discussion

SEOM is rarely reported in the literature. To our knowledge, the findings presented here represent the largest case series, involving 12 patients with restrictive tropia and SEOM. Mechanical restrictions or muscle palsy are usually responsible for diminished ocular motility. Common conditions involving congenital restrictive strabismus include CFEOM, Brown’s syndrome, inferior rectus fibrosis, and strabismus fixus [15]. SEOM exhibits a low morbidity worldwide, and the clinical manifestations are complicated and vary across geographic regions in terms of size, the morphology of the anomaly, and the extent of optic nerve involvement. SEOM was serendipitously identified as a result of autopsies and imaging, as its restrictive effects may not induce any symptoms or signs [10,16].

Some novel and interesting findings emerged from our study. First, the most prominent finding was that of the unilateral limitation of eye movement in more than three directions. The five patients older than 18 all had moderate to severe limited eye movement in at least three directions. In contrast, two of the seven patients in the younger group displayed this feature, indicating that the restriction developed with age and spread extensively to other EOMs, making the eye position more fixed. There were no significant relationships between the anatomy of the abnormal tissue and limited ocular motility in our cases. The more prominent restriction was found in the direction opposite to the location of the abnormal tissue attachment. Second, in our SEOM cases, there was a similar proportion of vertical and horizontal strabismus, and the degree of this strabismus did not appear to differ.

Third, in this review, we provided data on visual acuity, a measure which has received little attention in previous reports. In a case series as described by Molinari et al. [12], two out of seven cases exhibited worse vision (no less than two lines) in the affected versus the contralateral eye. Here, we found that 10/12 cases had worse visual acuity (≥two lines) in the affected eye, and the BCVA in five cases was <2/20. Eighty percent of the patients (Cases 3, 7, 10, and 12) with very large ocular misalignments were found to have very poor BCVA (lower than 2/20), while in the cases with an angle of deviation less than 40 PD, the proportion was 14% (only Case 2). Interestingly, the other case in the former group (Case 1, with SEOM in both eyes) had a BCVA of 12/20 and 20/20 (right and left eyes, respectively), suggesting that bilateral alternating fixations may avoid the suppression to a single eye from the cerebral cortex and help alleviate the extent of amblyopia [17]. It seems likely that in addition to strabismic amblyopia, ON dysplasia was also responsible for poor visual acuity. This suggestion is based on the imaging findings, which revealed a muscular slip between recti and the ON in Cases 11 and 12, with type 4 anomalous structures, and the observation that eight of our cases with other types of anomalies also showed ON involvement.

Finally, almost all patients in our review had one of following eyelid lesions: unequal palpebral aperture, incomplete eyelid closure, entropion, and/or trichiasis. In particular, a larger palpebral aperture in the affected versus contralateral eye may serve as a basis for a differential diagnosis of CFEOM, which is characterized by congenital ptosis [18].

Orbital imaging serves as an essential technique to confirm the diagnosis of SEOM. The classification of SEOM varies between investigators. The Lueder classification is the most commonly used. However, it is difficult to differentiate between the type 1 (additional muscle from other recti) and type 2 (anomalous fibrous tissues) categories by preoperative imaging. Both types were isotense to the rectus preoperatively, and so further methods, such as intraoperative observation and histopathological assessment, were essential [19]. The Demer classification is based on the direction of the involved recti. This method apparently fails to consider all types of SEOM, such as those originating from the Zinn ring.

In this study, we distinguished four types of SEOM within the 12 cases reviewed. These categories were based on the origins and insertions as observed from the imaging results. Type 1 SEOM involves a discrete accessory EOM inserting into the sclera, which is well-defined and tends to be readily identified with imaging. Therefore, there have been many reports describing this type of SEOM. Type 1 SEOM tends to insert into the posterior globe beneath the optic nerve, especially the inferotemporal quadrant [4,5,7,12]. Valmaggia [5] reported on the case of a 6-year-old boy with a similar anomaly and elevation deficiency in the left eye but no subjective complaints; thus, no strabismus surgery was performed. A case mentioned by Dobbs et al. [4] was diagnosed as Duane syndrome, and the patient underwent lateral rectus recession at the age of 23 months. Then the remaining limited supraduction and globe retraction indicated the existence of SEOM. The CT scan identified bilateral tissue isodense to muscle that was inferolateral to the optic nerve, and the postoperative pathological examination suggested that the tissue nature was similar to a normal EOM. Case 1 (Figure 1) displayed continuous exotropia after a previous strabismus surgery performed elsewhere (details unknown). We observed a 65 PD exotropia and 5 PD right hypertropia in the primary gaze with a moderate limitation of bilateral adduction and supraduction as well as a mild limitation of bilateral infraduciton in adduction. Coronal and axial CT scans revealed a bilateral symmetric accessory muscular tissue coursing from the annulus of Zinn to insert into the inferolateral sclera posteriorly. Interestingly, there was muscular calcification proximal to the globe in Case 1, a finding which has not been reported in previous studies. Compared with the other case types in this study, the clear structure of the SEOM in type 1 showed a smaller correlation with the optic nerve, indicating that it had less impact on visual acuity.

The second type of SEOM we identified was that wherein the rectus split into multiple bands, which then connected to other ERMs. From axial imaging, we found an abnormal band originating from the Zinn ring, which sometimes accompanied recti or formed connections with other recti as they entered the orbit. In such cases, multiple abnormal connecting bands among recti, deformed recti, and unclear boundaries were observed in the coronal views. From a recent anatomic study, the Zinn ring was found to be composed of the IR, medial rectus (MR), and lateral rectus (LR), with the SR confirmed to have an independent origin. It seems that muscular or tendinous connections tend to form between the SR and other recti, such as in the type 2 cases in this study [20,21]. In a cadaveric study of a 68-year-old male conducted by Haladaj [16], the SEOM originated from the annulus of Zinn laterally to the ON, then separated into two heads, which inserted into the superior rectus (SR) and inferior rectus (IR). In Case 2, imaging revealed a supernumerary orbital muscular band from the apex of orbit to other recti muscles, including the SR, IR, LR, and MR (Figure 2). This 4-year-old girl showed moderate to severe limitations in ocular motility in all directions except adduction. In addition, all the accessory structures of the three patients classified as type 2 seemed to be closely related with the ON, and the BCVA in the involved eyes was very poor (4/20, counting fingers and hand movement, respectively).

The third type of case consisted of an abnormal connection zone among the recti, with or without ON involvement. Compared with type 2, the type 3 cases were usually more distinct and identifiable. There was no significant difference in the probability of specific rectus involvement, with the five type 3 cases showing muscular bands connecting two or more recti, as shown in Table 1 and Figure 3. Khitri and Demer et al. [10] reported that abnormal vertical, horizontal, or varied directional connections, as well as levator–trochlear bands, were observed in their series of 12 cases. In the case series presented by Kightlinger et al. [13], an isointense thin tissue band coursing from the temporal edge of the SR to that of the IR was observed in all seven cases. As based on anatomical findings [16], the major vertical and horizontal anomalies observed appeared to involve duplications of the LR and SR muscles, a bifid insertion of the MR, and the absence of ERMs. Wide variations in ERMs and the directions of abnormal bands have been reported in previous studies. Overall, the SR appears to be more frequently involved [20,22,23].

The fourth classification type, with connections between recti muscles and the ON, has rarely been reported. No anomalous connections were observed between recti. Tetsu Naito et al. found that part of the structure of the SR and LR coursed from the sheath of the optic nerve or the oculomotor nerve [21]. The changes occurring during the short period of fetal development may remain rather than degenerate in the patients with additional tissue between the rectus and the optic nerve. In type 2 and type 3 cases, the optic nerve can also be surrounded by anomalous tissue (as in Case 2 in Figure 2), which may be associated with hypoplastic optic nerves. Cases 11 and 12 experienced amblyopia in the affected eye, with a mild fundus hypoplasia observed under ophthalmoscopic examination in Case 12.

The preoperative recognition of SEOM may enable a more personalized and effective surgical strategy. When the SEOM is located near the rectus muscle in the anterior part of the globe, an intraoperative exploration and resecting is recommended [3]. It is more feasible to resect the SEOM completely for the isolated tissue or adjacent to the rectus in type 1 cases. Dobbs attempted to apply a radiopaque string around the tissue and conduct CT scanning intraoperatively to ensure a complete excision [4]. However, anomalies that are situated deep within the orbit or have a close relationship with the optic nerve, especially those classified as type 2 and 4, sometimes make it difficult to perform a thorough resection. Under conditions wherein muscular bands cannot be lysed, the recession and resection of rectus muscles can serve as an effective therapeutic option to correct the strabismus. Among the seven patients treated with this procedure, surgical success was achieved in four cases and obvious improvement in two cases (Case 6 and Case 10). The appearance of Case 12 partly improved postoperatively. For Cases 7 and 10, which exhibited excessive ocular deviation, the intraoperative forced duction test results indicated that the restriction was so severe that it was impossible to perform a recession. Accordingly, muscle disinsertion was adopted as an alternative scheme, and globe fixation was also applied to strengthen the surgical correction. The angle of vertical deviation was 15 PD at the 2-month follow-up, but unfortunately recurred to 40 PD after 1 year in Case 7. For Case 1 in this study, the right MR was previously resected, and the new insertion was located 7 mm posterior to the nasal limbus; thus, LR recession 7 mm, MR resection 8 mm, and suturing 2 mm anteriorly onto the sclera OD were performed. Sixteen months after the second strabismus surgery, the ocular motility limitations were similar to the preoperative levels, and there was a slight improvement in adduction in the right eye; however, a residual 25 PD exotropia and substantial A pattern strabismus remained. We theorized that the bilateral SEOM may have exerted a restrictive force to displace both eyes outward and downward, resulting in a recurrent exotropia.

The current study had several limitations. Retrospective analyses are vulnerable to potential measurement inaccuracies and case selection bias. For example, the imaging performed in this study consisted of MRI and CT scans. The former is more effective for soft-tissue components, especially using high-resolution techniques, which would thus be the first choice for an accurate diagnosis of SEOM. CT scanning played a limited role, except in showing high-density structures, such as the calcified degeneration in SEOM observed in this study. A second consideration is that the SEOM anomalies were quite diverse, which resulted in complicated and varied clinical manifestations. In this way, the cases presented in this series did not represent the whole spectrum of SEOM. The low morbidity of SEOM makes it difficult to obtain a large sample size and thoroughly analyze the characteristics of the different types of SEOM, which then restricts the generalizability and applicability of results. The location of the SEOM deep within the orbit prevented us from obtaining SEOM samples during the strabismus surgery. As a result, the diagnosis of SEOM was based entirely on the imaging features and lacked a pathological confirmation. Finally, genetic testing was not performed in any of our cases.

## 5. Conclusions

For patients with atypical restrictive strabismus, imaging is strongly recommended to assess the possibility of SEOM. We identified different forms of SEOM, including the connection of recti, the anomalous tissue originating from the annulus of Zinn and inserting into recti or the globe, and the connection of recti with ON involvement. The limitation of ocular motility in ≥3 directions, worse visual acuity in the affected eye, and eyelid abnormalities (in particular, unequal palpebral aperture) were recommended as factors for the differential diagnosis of SEOM. Rectus recession with or without resection can serve as an effective surgical option when a complete excision of the SEOM is difficult to achieve.

## Figures and Tables

**Figure 1 medicina-58-01691-f001:**
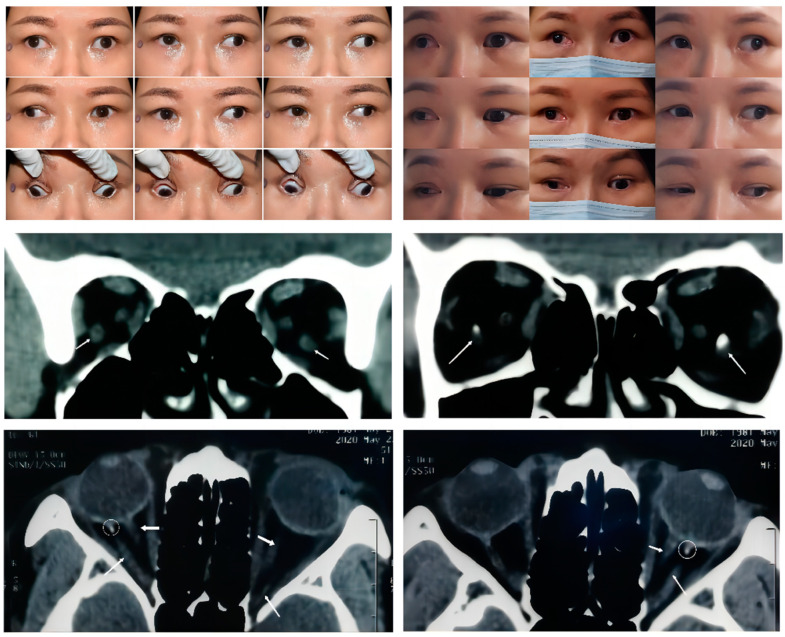
**Upper panel**: Nine gaze photographs of Case 1 showing a preoperative 65 PD alternating exotropia and a large A pattern strabismus. A moderate bilateral limitation of adduction and supraduction and mild bilateral limitation of infraduction were present in this case (left). Sixteen months after a second strabismus surgery (right), there was a slight improvement in the adduction of the right eye. A residual 25 PD exotropia and large A pattern strabismus remained. **Middle Panel**: Arrow in the coronal CT scan indicates a bilateral symmetric accessory extraocular muscle. The high-density image within the muscle suggests a local calcification just behind the posterior pole of the eyeball (the right picture). **Bottom Panel**: Thin arrow in the axial CT scan indicates that the accessory extraocular muscle in both eyes originat from the apex of the orbit and inserting into the posterior globe lies between the lateral rectus and optic nerve. Thick arrow: the optic nerve. Circle: local calcification within the muscle.

**Figure 2 medicina-58-01691-f002:**
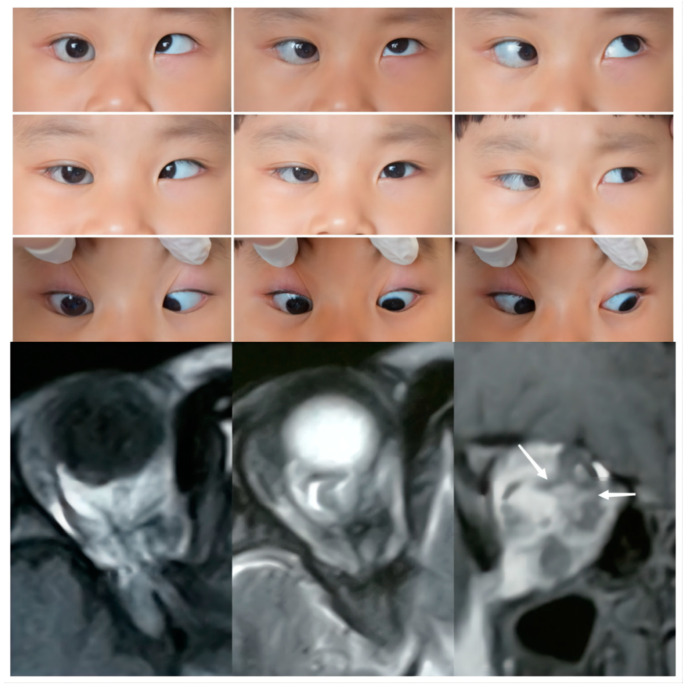
**Upper Panel**: Nine gaze photographs of Case 2 showing a 30 PD esotropia and -4 limitation in the abduction of the right eye. The right eye showed eyelid entropion, with palpebral aperture 2 mm larger than that of the left eye. **Bottom Panel**: T1 and T2 axial and a T2 coronal MRI of Case 2. Anomalies in EOM bands extended from the apex of orbit and connected with multiple recti. The right optic nerve was maldeveloped and its path was unclear.

**Figure 3 medicina-58-01691-f003:**
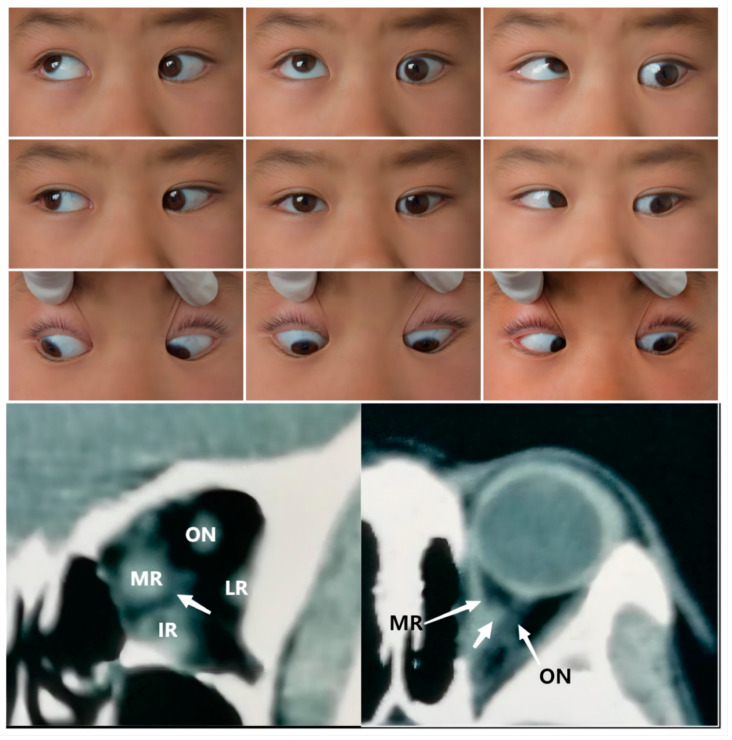
**Top Panel:** Nine gaze photograph of Case 8 showing a 35 PD of left hypotropia, -3 limitation in OS abduction, and -4 limitation in OS supraduction. The left inferior lid margin was 2 mm lower than that within the right eye. **Bottom Panel**: Coronal (left) and axial CT of Case 9 showing a unilateral anomaly of EOM bands connecting medial rectus (MR) and inferior rectus (IR) in the left eye, as well as muscular slip between the medial rectus and optic nerve (ON). IR: inferior rectus; LR: lateral rectus.

**Figure 4 medicina-58-01691-f004:**
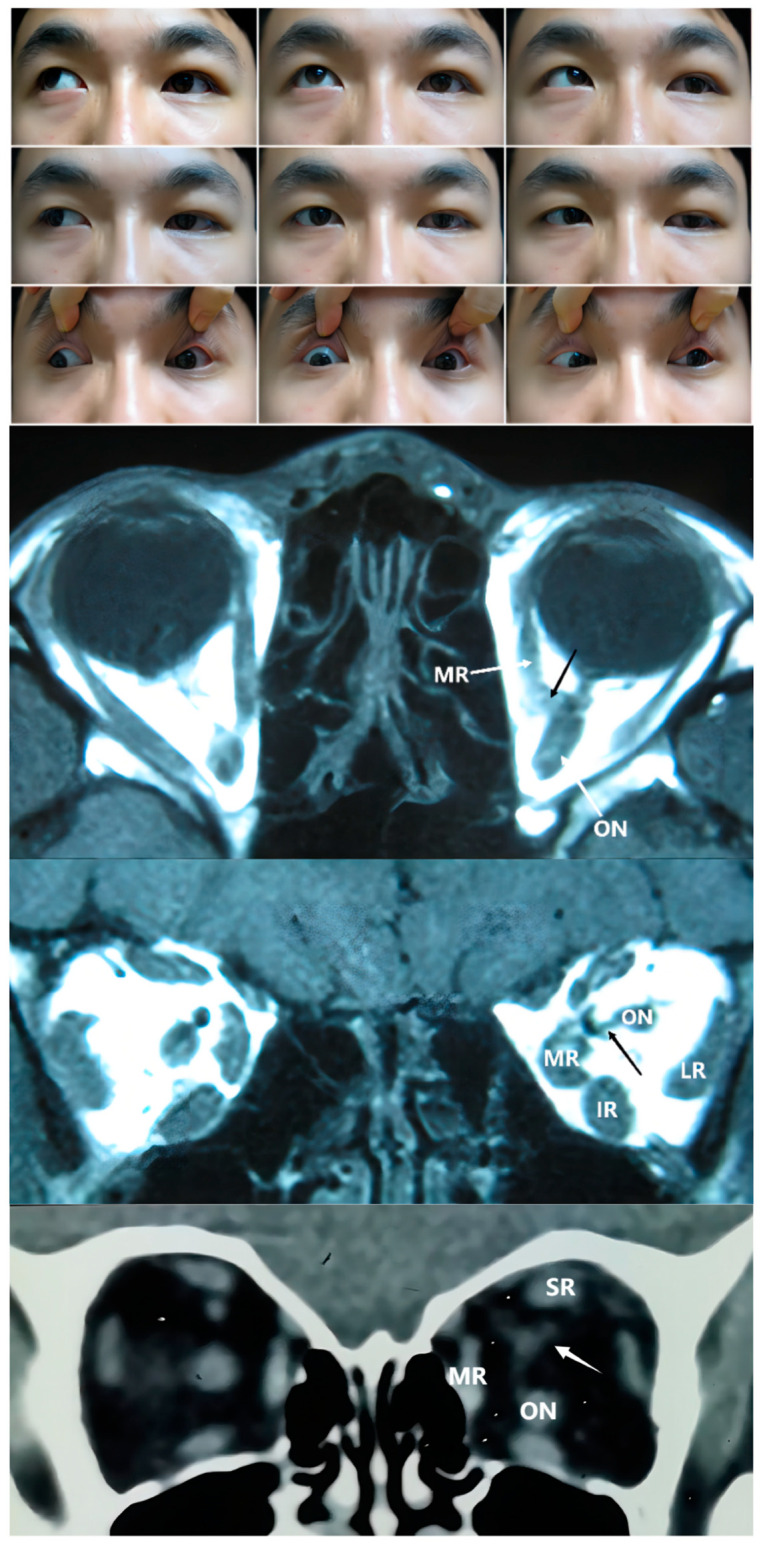
**Upper Panel:** Nine gaze photograph of Case 11 showing a 13 PD exotropia and 13 PD left hypotropia. Ocular motility determinations revealed differences in the extent of limitations in all directions. **Second Panel:** T1 coronal MRI of Case 11 demonstrating a muscular band between the medial rectus (MR) and optic nerve (ON) in the left orbit. **Third Panel**: T1 axial MRI of Case 11 showing connections. **Bottom Panel:** coronal CT of Case 12 showing a muscular band between the superior rectus (SR) and optic nerve in the left orbit. IR: inferior rectus; LR: lateral rectus.

**Figure 5 medicina-58-01691-f005:**
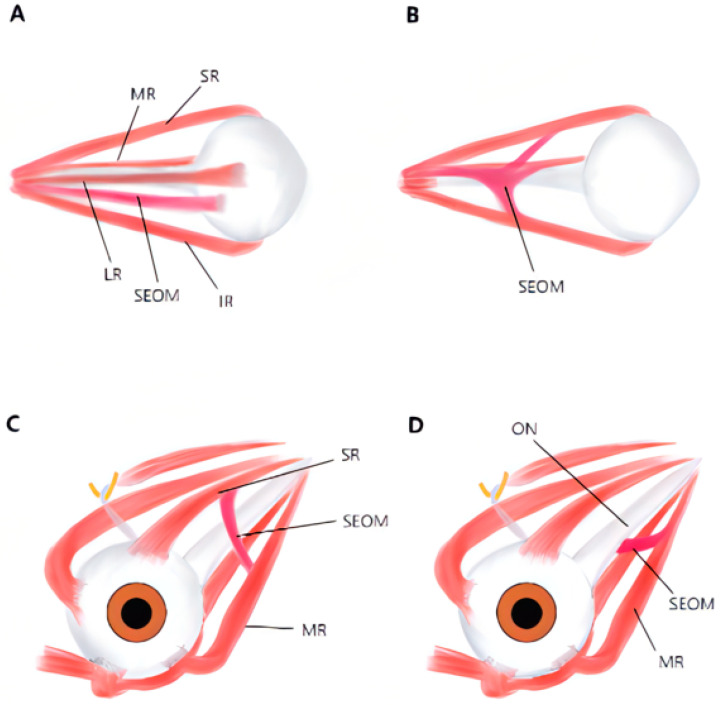
Diagram showing four types of supernumerary extraocular muscles (SEOMs): (**A**) type 1 coursing from orbital apex to posterior part of globe; (**B**) type 2 from Zinn annulus to other recti; (**C**) type 3 connecting recti with or without optic nerve (ON) involvement; (**D**) type 4 only connecting rectus and ON. SR: superior rectus; IR: inferior rectus; LR: lateral rectus; MR: medial rectus.

**Table 1 medicina-58-01691-t001:** Clinical characteristics and surgical results in 12 patients with supernumerary extraocular muscle.

Type	Case	Age	Sex	Side	BCVA	Preoperative Deviation (PD)	Preoperative Motility	SEOM Location in MRI/CT	ON Involvement	Surgery	Follow-Up (Month)	Postoperative Deviation (PD)	Postoperative Motility (Affected Eye)
1	1	38	F	OU	OD: 12/20 OS: 20/20	XT = 65 R/L = 5 A-pattern	OD 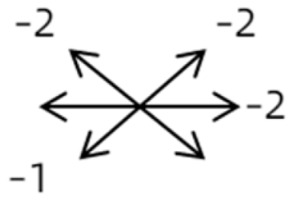 OS 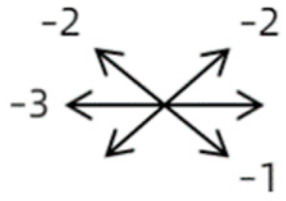	Zinn ring to the posterior globe	No	1st: unknown 2nd: RLR rec 7 mm, RMR res 8 mm + adv 2 mm	16	XT = 25 A-pattern	OD 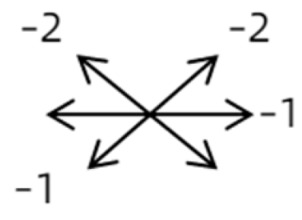 OS 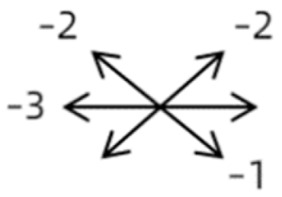
2	2	4	F	OD	OD: FC OS: 16/20	ET = 30	OD 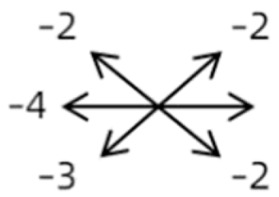 OS 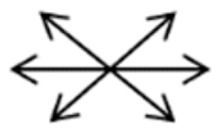	SR to multiple recti	Yes	RMR rec 6 mm	30	ET = 3	OD 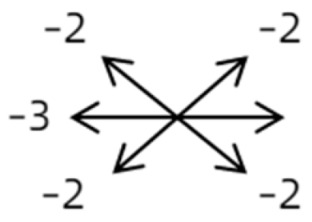 OS 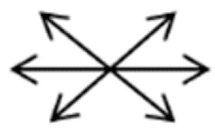
2	3	4	F	OD	OD: HM OS: 10/20	XT = 85	OD 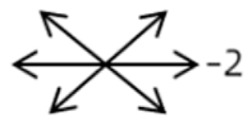 OS 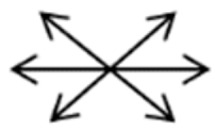	Additional head of LR to SR	Yes	1st: RLR rec 9 mm + RMR res 14 mm 2nd: LLR rec 6 mm	48	XT = 5	OD 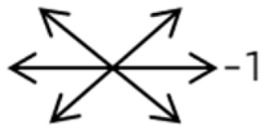 OS 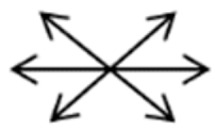
2	4	6	F	OD	OD: 4/20 OS: 20/20	R/L = 13	OD 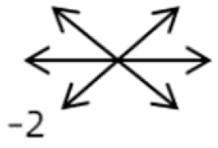 OS 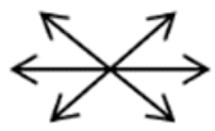	SR to MR	Yes	No (refused)			
3	5	23	F	OD	OD: 16/20 OD: 20/20	ET = 26	OD 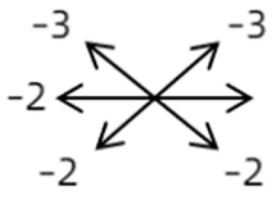 OS 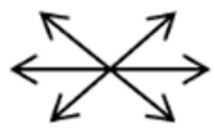	IR to MR	Yes	No (refused)			
3	6	8	M	OD	OD: 20/20 OS: 20/20	L/R = 18	OD 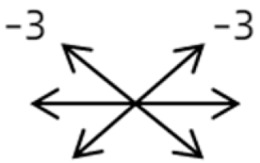 OS 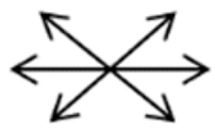	IR to MR	No	RIR rec 7 mm + globe fixation	60	L/R = 8	OD 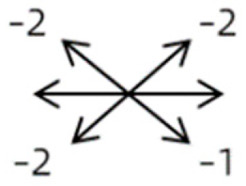 OS 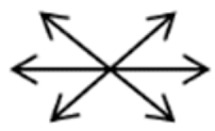
3	7	24	M	OS	OD: 20/20 OS: LP	L/R > 80 XT = 12	OD 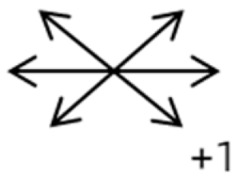 OS 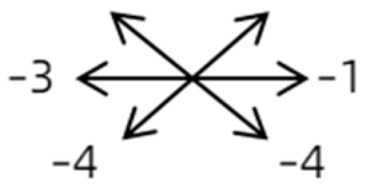	SR to LR, MR	Yes	LSR dis + LSO dis + globe fixation	12	XT = 15 L/R = 40	OD 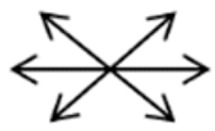 OS 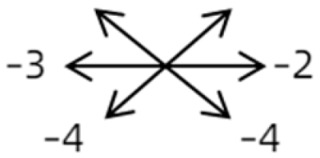
3	8	9	M	OS	OD: 20/20 OS: 14/20	ET = 2 R/L = 35	OD 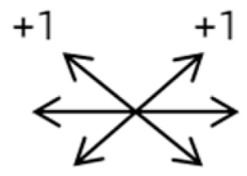 OS 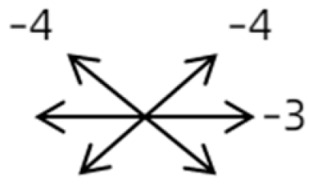	MR to IR	Yes	LIR rec 5 mm	30	R/L = 2	OD 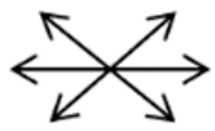 OS 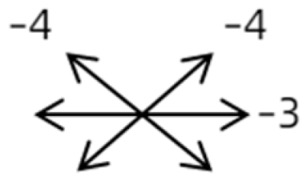
3	9	8	M	OS	OD: 20/20 OS: 2/20	XT = 10 R/L = 40	OD 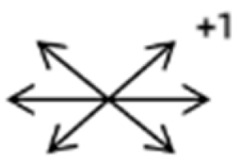 OS 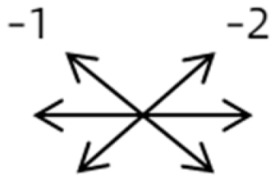	MR to LR, IR	Yes	LIR rec 5 mm + LSR res 5 mm + LSO tenotomy	36	R/L = 3	OD 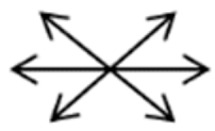 OS 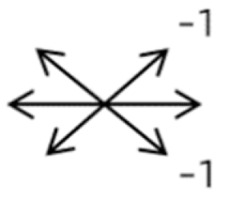
3	10	25	F	OD	OD: FC OS: 20/20	ET > 65 L/R > 65	OD 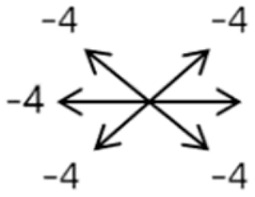 OS 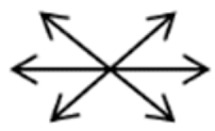	MR to LR	Yes	RMR dis + globe fixation	6	ET = 10 L/R = 20	OD 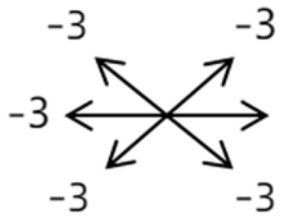 OS 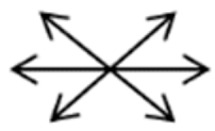
4	11	18	M	OS	OD: 20/20 OS: 8/20	XT = 13 R/L = 13	OD 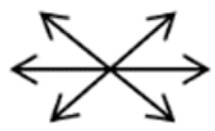 OS 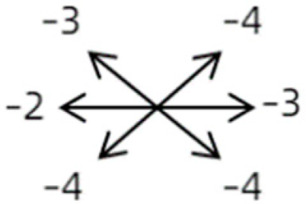	MR to ON	Yes	No			
4	12	4	F	OS	OD: 14/20 OS: 14/20	XT = 18 L/R = 64	OD 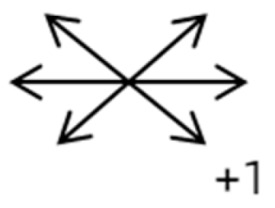 OS 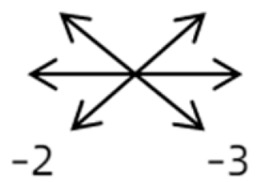	SR to ON	Yes	LSR rec 7 mm	12	L/R = 30	OD 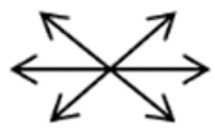 OS 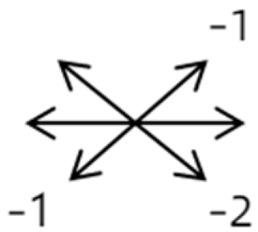

F: female, M: male; OD: right eye; OS: left eye; FC: counting finger; HM: handmoving; XT: exotropia; ET: esotropia; L/R: right hypotropia; R/L: left hypotropia; LP: light perception; SR: superior rectus; IR: inferior rectus; LR: lateral rectus; MR: medial rectus; SO: superior oblique; ON: optic nerve (involvement); rec: recession; res: resection, adv: advancement; dis: disinsertion.

**Table 2 medicina-58-01691-t002:** Incidence of signs and symptoms.

Characteristic	Number of Cases	Percentage (%)
Ipsilateral side: worse BCVA (≥2 lines)	10	83.33
High hyperopia	3	25.00
Unequal palpebral aperture (≥2 mm)	10	83.33
Incomplete eyelid closure	3	25.00
Entropion and trichiasis	4	33.33
Enophthalmos	3	25.00
Dysplastic optic nerves in fundus photograph	3	25.00
Very large horizontal strabismus	3	25.00
Very large vertical strabismus	3	25.00
Elevation deficit in abduction	8	66.67
Limited ocular motility in multiple directions (≥3)	7	58.33
Optic nerve involvement in imaging examination	10	83.33

**Table 3 medicina-58-01691-t003:** Pre- and postoperative outcomes of patients who underwent surgery (n = 9).

Study Parameter	Preoperatively, Mean ± SD (IQR)	Postoperatively, Mean ± SD (IQR)	*p* Value
Horizontal deviation (PD)	33.6 ± 32.0 (10, 65)	6.4 ± 8.2 (0, 10)	0.040
Vertical deviation (PD)	37.2 ± 29.4 (13, 64)	11.4 ± 14.1 (0, 20)	0.056
Supraduction restriction	4.1 ± 3.3 (0, 8)	3.1 ± 2.6 (1, 4)	0.652
Infraduction restriction	3.1 ± 3.2 (0, 5)	3.2 ± 2.3 (2, 4)	0.686
Horizontal restriction	2.4 ± 1.9 (0, 4)	1.8 ± 1.9 (0, 3)	0.491
Total restriction of duction	9.7 ± 5.7 (5, 13)	8.1 ± 5.0 (4, 13)	0.658

PD, prism diopters; SD, standard deviation; IQR, interquartile range; Supraduction restriction: sum of restrictions of lateral and medial supraduction. Infraduction restriction: sum of restrictions of lateral and medial infraduction. Horizontal restriction: sum of restrictions of abduction and adduction. Total restriction of duction: sum of restrictions of all directions. *p*-value: Wilcoxon signed-rank test.

## Data Availability

All data generated or analyzed during this study are included in the article. Further enquiries can be directed to the corresponding authors.

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
