# Peer review of "Supernumerary Extraocular Muscle: A Rare Cause of Atypical Restrictive Strabismus"

_medicina, 2022, doi:10.3390/medicina58111691_

Round 1

Reviewer 1 Report

The authors reported an interesting case series concerning a retrospective analysis of patients diagnosed with supernumerary extraocular muscles (SEOM) and restrictive strabismus using orbital imaging and suggested a novel anatomical classification of SEOM. The followings are some comments regarding the manuscript.

1. In the introduction, the author quoted Khitri and Demer's results, however, there was actually no statistical significance between the two groups. The authors should rephrase the sentences. 

2. Since the authors suggested a new classification for SEOM, the authors should organize their cases in the proposed classification in a more followable sequence for the readers to track and understand their results throughout the manuscript, especially in Table 1.

3. In Table 1, the author did not specify how much recession was done on the right inferior rectus muscle in case 5; and what does globe fixation mean? The same concern was also raised for case 7 and 12, does "globe fixation" mean the detached muscles were fixated to the original insertion or other structure in the orbit?

4. In Table 1, the ocular motility was presented using a diagram but only for the SEOM involved side. The author should present both eyes for the readers to better analyze the overall ocular motility. And is there any overaction in the contralateral eye among the case series?

5. For the case demonstration of their proposed new classification of SEOM, the author should reconsider the orbital image for a more identifiable SEOM in each category. The image size and presentation of the demonstration photos should be organized better. 

6. Since the authors mentioned the presented visual acuity and related poor vision in this case series (which is also listed as one of the novel findings of this research), they should discuss more on this topic and the relationship with SEOM or identify the possible cause of poor vision (or amblyopia) in each case. In addition, the visual acuity of both eyes would also be important information to be included. 

7. In their new classification of type 3 and 4, the authors mentioned optic nerve involvement; and in type 4 cases, optic nerve hypoplasia was also mentioned. Besides, in Table 1, cases classified as type 1 and 2 have also been listed with positive optic nerve involvements. Could optic nerve involvement, aside from the proposed classification, be a more determining factor affecting visual acuity?

8. The author reported 6 cases with successful surgical outcomes. However, according to their definition (ocular deviation in the primary gaze of  10 PD in horizontal and  5 PD in vertical directions), there are only 4 cases being counted as a success. And since there was no statistical significance between the pre-and postoperative results, the author should not conclude that any strabismus procedure can be used successfully in these SEOM patients, especially when the differences between the individual pathology in their case series were considerable.

9. As suggested by the author, the hypothesis for SEOM is still controversial in the literature, in a recent embryological study, "Examination of the Annular Tendon (Annulus of Zinn) as a Common Origin of the Extraocular Rectus Muscles: 2. Embryological Basis of Extraocular Muscles Anomalies", there could be some anatomical evidence for different types of SEOM. The author should include this article in their discussion for a deeper discussion of their proposed new anatomical classification and rethink if there should be a new nomenclature of the SEOM disease spectrum for different embryological origins of different types of SEOM.
10. Some paragraphs in the discussion should be included in the introduction or considered redundant. 
11. Please recheck and modify the abbreviation and reference format to conform to the journal's style.

Reviewer 2 Report

To the authors to present the “Supernumerary Extraocular Muscle: A Rare Cause of Atypical Restrictive strabismus”

We are facing a well-worked and presented paper, I dare to suggest some changes:

#1 There are descriptions in the discussion that could be included in the introduction.

#2 The discussion is very long, try to shorten it a bit.

#3 The description of the figures is very long, try to be more concise.

#4 A paragraph that differentiated the results by age could be included in the discussion, there is a lot of dispersion in age.

#5 The conclusion should be more concise, with fewer assumptions, it seems to leave the results in the air, and should be supported in them.

Reviewer 3 Report

The authors describe "Supernumerary extraocular muscle (SEOM) belongs to an extremely uncommon entity in which more than six extraocular muscles (EOM) are present and can result in restrictive motility and strabismus."

They reported the extreme rare conditions well. And written English is acceptable. The study design and results looks like technically sound and meaningful to clinical science.

In advance of review, thank  you for great review and valuable data processing. Although it is low interest to the general readers and ophthalmologists, originality and novelty looks quite high level.

I do not feel any further comments to this manuscript such as presentation, scientific soundness except extreme rare conditions.

Thank you again for valuable research.

Round 2

Reviewer 1 Report

Thank you for the authors’ attentive responses. There are some follow-up questions regarding the updated manuscript.

1.Despite the confusion raised by the automatic pdf formatting done to the extensively modified new Table 1, some points should be clarified.

i.In cases 5, 7, 8, and 11, under the column of “ON involvement”, there are answers other than ‘yes’ and ‘no’ (ex: MR and ON, etc.), what does this mean? Please clarify.

ii.In the new Table 1, the column of “preoperative motility” has been completed by adding a binocular diagram to establish a more comprehensive viewpoint for the readers. However, why does the author omit to present the binocular status of postoperative motility in the last column?

iii.In case 10 (possible case 12 in old Table 1), the preoperative deviation differed from the original data. Please clarify the reason for this difference.

2.Since the author proposed a new anatomic classification for supernumerary extraocular muscles (SEOM) and because of the rarity of this disease entity, the case number is considered to be low for a classification being suggested (especially in type 1). The author should include a literature review that could further expand the case number for their new anatomic classification. Besides, it would be much more useful if a suggested classification has a clinical implication, for example, a restrictive pattern or visual prognosis of a certain type differs from the other in the given classification. The authors should elaborate more on this topic.

3.As mentioned in the previous comment, the orbital image examples of each classification type were not in their optimal quality for demonstration. The author should reconsider using other more demonstrable images or other cases in each classification. Or maybe, the authors could try using drawings to illustrate their proposed classification for a clearer demonstration to the readers.
